# CNNGen: A Generator and Benchmark for Sustainable Convolutional Neural Network Search

## Abstract

Neural Architecture Search (NAS) emerged as a promising approach to search for optimal neural network architectures in a limited, predefined architecture space. One popular method to form such a space is to derive a known architecture in which we insert *cells* where NAS algorithms can automatically combine network functions and connections. Cell-based methods yielded hundreds of thousands of trained architectures whose specifications and performance are available to design performance prediction models. Cell-based approaches come with three main limitations: *i)* generated networks have limited diversity resulting in very similar performances, in turn hampering the generalization of trained performance models, *ii)* networks' implementations are missing hampering performance understanding, and *iii)* they solely focus on performance metrics (*e.g.*, accuracy) ignoring the growing sustainability concern. We propose CNNGen, an approach that addresses: *i)* by leveraging a domain-specific language (DSL) to automatically generate convolutional neural networks (CNNs) without predefined cells or base skeleton. It allows the exploration of diverse and potentially unknown topologies; *ii)* CNNGen's comprehensive pipeline stores the network description (textual and image representation) and the fully executable generated Python code (integrated with popular deep-learning frameworks) for analysis or retraining, and *iii)* in addition to training and performance metrics, CNNGen also computes energy consumption and carbon impact for green machine learning endeavors. We demonstrate the possibilities of CNNGen by designing two performance predictors and comparing them to the state of the art.

## 1 Introduction

Deep learning algorithms now efficiently power a wide range of applications, including image and speech recognition (Lecun et al., 1998; Hinton et al., 2012), natural language processing (Touvron et al., 2023), and autonomous vehicles (Nesti et al., 2023). Given the numerous neural architectures available nowadays, finding the architecture that will perform best for a specific task is tedious and requires expert knowledge (configuration of the architecture, including functions to apply). To address this, Neural Architecture Search (NAS) (Elsken et al., 2017; 2019; Jin et al., 2019) has emerged to automate the exploration and design of neural architectures by solving an optimization problem. To support this research, the community developed benchmarks such as NASBench101 (Ying et al., 2019). These benchmarks derive thousands ($\approx$ 423K for NASBench101) of unique architectures from a skeleton by combining connections and network functions in predefined cells. Though the availability of such datasets is crucial for NAS optimization algorithms, they come with key limitations. First, generated architectures lack diversity: 99% of generated NASBench101architectures have a training accuracy on CIFAR-10 above 99% when trained for 108 epochs. In such cases, it is difficult for performance prediction or classification models to discriminate highly performing from badly performing ones. Second, essential artifacts (such as the fully trainable model code) are missing, limiting reproducibility. Finally, these datasets tend to ignore the computational and environmental costs incurred by training all these architectures by not recording them in the resulting datasets. Recent studies show that these costs cannot be overlooked anymore (Wu et al., 2022; Patterson et al., 2021). For example, LLaMA (Touvron et al., 2023), a large language model, required

$2,048$ high-end GPUs for an estimated carbon footprint of 173 tCO2eq[1]. Another study estimated that the training of a single deep learning model can emit as much carbon as five cars over their lifetime (Strubell et al., 2019). Although we target smaller architectures here, training thousands of them may not be possible for smaller structures and may also result in carbon emissions.

In this paper, we propose CNNGen, *a different approach for generating convolutional neural networks (CNNs)*. CNNGen relies on a domain-specific language (DSL), capturing the architecture space via a dedicated grammar. Rather than systematically varying topologies in fixed cells, it constructs CNNs from scratch, assembling blocks according to grammar rules (as the example shown in Figure 2). It results in a much more diverse set of network topologies. We randomly generated about $1,300$ CNNs having from 9 to 272 layers and we observed widespread performance metrics. We also store the full model code, the image representation (*i.e.,* a PNG image as shown in Appendix A) of the network, and the computational and carbon footprint for training each network. To demonstrate CNNGen relevance, we built two performance prediction models able to use the PNG images of the networks and their code and compare them to a state-of-art performance predictor (Wen et al., 2019). Our code-based predictor outperforms the two others, highlighting the interest in having this information while predicting performance. Our contributions are as follows: i) a specialized DSL explicitly tailored for defining neural architectures; ii) a DSL-driven architecture generator, transforming high-level architecture descriptions into executable Python code; iii) a diverse and extensive dataset of neural architectures including their detailed architecture descriptions, corresponding Python code, PNG images of the CNN structures, and crucial metrics like accuracy and energy consumption for reusability; iv) two novel performance predictors, leveraging the generated Python code and CNN images.

Section 2 presents key concepts regarding NAS and DSLs. Section 3 motivates our different approaches to CNN generation and why the use of a DSL is appropriate. Section 4 describes our DSL and how we leverage it to build automatically CNNs, and our dataset. Section 5 presents the predictors we used in our evaluation. Section 6 use our set of neural architectures generated with our DSL to evaluate our predictors. Section 7 wraps with conclusions and presents future work.

**Open science policy.** Our companion website comprises[2]: the CNNGen tool, the dataset of $1,300$ generated architectures and their performance metrics. It also comprises the performance prediction models used in our evaluations.

## 2 BACKGROUND

**Neural Architecture Search** (NAS) focuses on automating the design of neural network architectures, focusing on search space, search strategy, and performance estimation. Various search strategies exist including random search, Bayesian optimization, evolutionary methods, and reinforcement learning (Zoph & Le, 2017; Real et al., 2019; Miikkulainen et al., 2019). Performance estimation techniques, such as learning curve extrapolation or network morphisms, aim to reduce computational costs (Cai et al., 2018; Domhan et al., 2015; Jin et al., 2019). AutoML, exemplified by Auto-Keras (Jin et al., 2019; 2023), simplifies machine learning model usage for individuals with limited expertise. While Auto-Keras starts with an existing architecture and modifies it through morphing, it requires a good starting point and explores a small neighbourhood around the current architecture. Removing these constraints may allow for a more extensive exploration of the possible architectures, in turn, making the search for optimal architectures more interesting.

**Domain-Specific Language** (DSL) is a high-level software language that captures the essence of a particular domain via specialized abstractions (Wąsowski & Berger, 2023). A DSL typically provides a concise, expressive syntax that enables developers to easily and accurately enounce complex ideas. It can also capture domain knowledge with constraints, reducing the risk of errors or inconsistencies. Domain-specific languages exist for numerous domains (*e.g.,* software engineering, biology, or robotics) and at various abstraction levels (from programming GPUs to language modelling). In the context of neural architecture search, using a DSL can be beneficial for several reasons.

---

[1]The term "tCO2eq" means a metric measure used to compare the emissions from various greenhouse gases on the basis of their global-warming potential.

[2]https://doi.org/10.5281/zenodo.8387493

*First*, the task of neural architecture search is highly complex, posing significant challenges when developing efficient algorithms with general-purpose programming languages (as the usual strategies in current ML communities). The DSL can mitigate this from a new perspective, simplifying the representation of neural architectures, and making it easier to explore the search space efficiently. *Second*, a DSL can ease the creation of specialized tools and workflows tailored to the needs of NAS researchers and practitioners. For example, a DSL can automate recurrent tasks such as data preprocessing, model training, and hyperparameter tuning, which can save time and reduce the risk of errors. *Third*, the DSL can promote standardization and collaboration within the neural architecture search community. By providing a common language and set of tools, a DSL enables researchers and practitioners to share and reproduce experiments, which can help accelerate progress and improve the overall quality of research in the field. In conclusion, using domain-specific language in neural architecture search tasks enhances efficiency, accuracy, and collaboration, accelerating progress toward more effective and efficient neural network designs.

## 3 MOTIVATION

Ying et al. (2019) built a reusable benchmark dataset (NASBench101) that uses a specific neural network topology. It is defined as a convolution layer, followed by what is called a "stack" and then a downsampling layer, another stack, another downsampling layer, a third stack, a global average layer, and finally a dense layer. Stacks are a succession of three cells and each cell can contain a maximum of 9 transitions (from one layer to another) and/or 7 layers. Layers in a cell can only be $3 \times 3$ convolutions, $1 \times 1$ convolutions, or max-pooling. Any combination (with repetitions) of these layers is valid as long as the number of maximum layers and transitions between them is observed. In the end, NASBench101 encompasses approximately $423,000$ unique graphs and provides a dataset of performances when they are trained on CIFAR-10. Yet, plotting the performances of all the architectures shows that there are little differences in the observed accuracies as shown in Figure 1a. The distribution of the accuracy reported from the test set shows that the queue of the distribution gathers the majority of the models. They can achieve $80\%$ accuracy and above. We attribute this small variance in performance to a limited search space around a predefined architecture. We would like to express those architectures but also to explore more freely their possible combinations. To do so, we introduce a grammar for CNNs to sequence layers composing an architecture (see Section 4.1). On the contrary, Figure 1b shows that the reported distribution of CNNGen models accuracy peaks at about $55\%$ and normal Gaussian form of the distribution spread between $40\%$ and $68\%$.

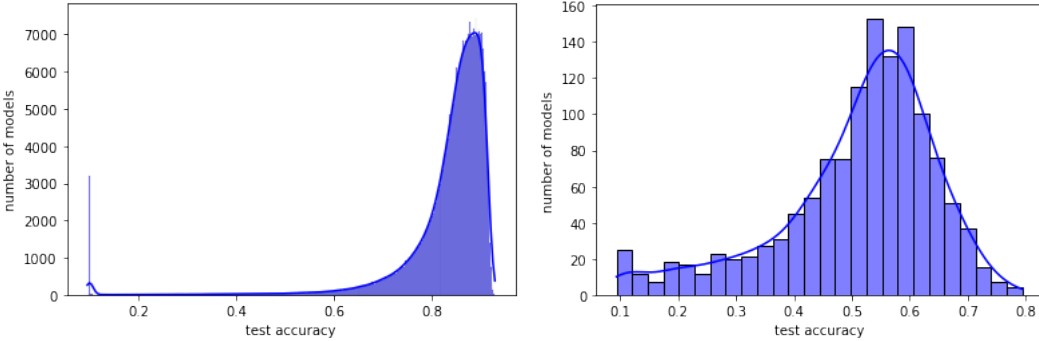

(a) Distribution of accuracies of NASBench101 models on CIFAR-10 test set after training for 36 epochs.

(b) Distribution of accuracies of $1,300$ models generated with CNNGen on CIFAR-10 test set after training for 36 epochs.

Figure 1: Comparison of the distribution of the accuracies of CNN fromNASBench101and from CNNGen when evaluated on CIFAR-10 after a training for 36 epochs. NASBench101models provide a higher accuracy (mean value of $82\%$ while CNNGen's mean value is $55\%$). Yet, CNNGen's boxplot spread wider suggesting there is potential to look for models that perform the best.

In addition, we take into account not only the classification performance but also the energy consumption that is merely considered by other works. While climate change is now undeniable and we know we should reduce and limit our resource consumption, the size of CNN architectures keeps

increasing (Bernstein et al., 2021) suggesting that more data are needed for training, more layers compose the architectures and thus more operations are performed when running the CNNs. However, adding layers is not necessarily a guarantee for higher performance and it may be difficult to estimate the ideal size of a network. Thus, we would like to offer tradeoffs between the size and other characteristics such as the computational cost and related carbon impact.

## 4 CNN Generator (CNNGen)

CNNgen generates CNN architectures thanks to a context-free grammar framework (Bettini, 2016; Combemale et al., 2016). This grammar describes the topology (*i.e.,* the structure of the topology) of a CNN. The grammar consists of tokens that relate to layers in a CNN and rules, extracted from the domain, constrain their sequencing. CNNGen provides an editor that allows specifying the CNN topology and can transform automatically this specification into the model's code using the Keras Python framework.

### 4.1 Specifying Neural Architectures

CNNGen relies on Xtext (Bettini, 2016; Combemale et al., 2016) for its context-free grammar framework. Xtext is supported by the Eclipse Foundation and offers a complete infrastructure covering editors, parsing, compilation, and interpretation. Formally, a grammar consists of a finite set of non-terminals, a finite set of terminals, and a finite set of grammar rules. Our analysis of different CNN architectures (*e.g.,* LeNet, AlexNet, and ResNet among a few others), led us to extract five key concepts used to describe architectures. Figure 2 shows the different rules that we defined for these concepts and terminals in our grammar. The full grammar is provided in Appendix B.

| | | |
|---|---|---|
| `architecture` | $\rightarrow$ | input featureExtraction* featureDescription classification* output |
| `featureExtraction` | $\rightarrow$ | conv \| BatchNormalisation \| pooling \| skipConnection |
| `featureDescription` | $\rightarrow$ | flatten \| globalPooling |
| `classification` | $\rightarrow$ | fullyConnected |

Figure 2: CNNGen grammar rules

The first row shows how the concepts are combined to form a valid CNN architecture. It reads as follows: a CNN architecture must be composed of an input layer, followed by one or multiple layers for feature extraction, then a layer performs what we call feature description. Then, there must be one or multiple classification layers, and finally a single output layer. The three remaining rows specified what are the literals for each of the other keywords (*i.e.,* featureExtraction, featureDescription, and classification). For instance, valid layers for featureDescription are "flatten" or "globalPooling". For now, a classification layer can only be fully connected. We can easily extend this grammar to support additional constructs or constraints.

As an example, let us show how we can describe a LeNet5 architecture. Figure 3 shows a graphical representation of a LeNet5 as depicted in the original paper Lecun et al. (1998). This CNN is composed of an input layer, followed by a convolution layer, a subsampling (now called pooling) layer, a second convolution layer and a second subsampling layer, a flatten layer, and three fully connected layers before the output. The LeNet5 architecture can be described in CNNGen as shown in Figure 4. In the following, we show how to generate the model's code from a CNN specification.

### 4.2 CNN Generation Process

We rely on Xtend (Bettini, 2016) to generate the code of the trainable model. Regarding the topology, the translation is straightforward since our grammar terminals map to Keras' constructs. Regarding the hyperparameters, we need to handle multiple aspects. For instance, when dealing with a convolution layer, we need to set the kernel size, the stride, the activation function, *etc.* When it comes to a drop-out layer, we need to set the drop-out rate, and it goes on for every kind of layer.

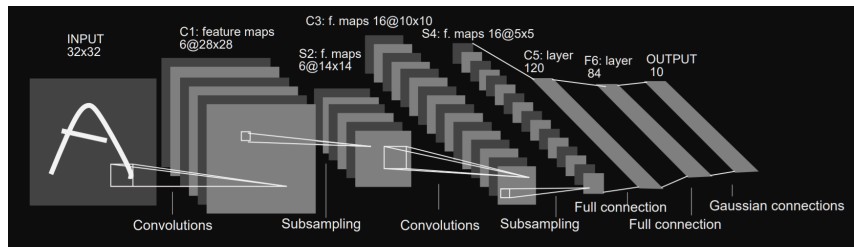

Figure 3: LeNet-5 architecture as described in Lecun et al. (1998)

LENET $\rightarrow$ input conv pooling conv pooling flatten fullyConnected fullyConnected output

Figure 4: LeNet5 described in the language defined in CNNGen

Since our grammar allows combinations of existing topologies, we do not have any prior about hyperparameter values that would maximize performance. therefore, we set them to random values. Hyperparameter values are not independent, ignoring this will likely yield non-trainable models. For instance, in a residual block of a ResNet, when the residual connection is merged back to the backbone, the merge is possible only if the two parts of the connection have the same feature dimensions (this requirement comes from Python that is not able to cope with the merge of two structures of different sizes). Our generator takes into account such rules to minimize the number of non-trainable models (less than 1% of our generated networks were not trainable). Future evolutions of CNNGen will see smarter strategies for hyperparameter selection and a DSL allowing experts to specify them.

### 4.3 CNNGEN OUTPUTS

We implemented a random generator in CNNGen so that it can serve as a benchmark generator. In that case, the production and training of CNNs is fully automatic, and this is how we generated our dataset of 1,300 networks. As for the metrics, we record the following ones: accuracy, training time in seconds, epochs executed, and training parameters, but also Emissions (CO2-equivalents [CO2eq], in kg), Emissions Rate (Kg/s), CPU Power (W), GPU Power (W), RAM Power (W), CPU Energy (Kw), GPU Energy (Kw), RAM Energy (Kw), and Total Energy Consumed (Kw), and location-specific data such as Country Name / ISO Code. Here we use the Code Carbon [3] library to calculate measurements related to energy consumption. To ensure reproducibility, NAS-Bench101 (Ying et al., 2019) made available the adjacency matrix of the different topologies. This way, the different topologies that are in the benchmarks can be reproduced. Yet, this cannot ensure the reproducibility of the results completely as, for instance, the hyperparameters may differ. To improve this aspect, we make public, for a single model, the description of its topology in our DSL, the generated Python code, a PNG image representing the model (*i.e.,* the topology with hyperparameters), and associated measures.

### 5 PERFORMANCE PREDICTORS

NAS is a field that aims at reducing the cost of training ML models in the long run. It starts with the training and evaluation of a set of ML models and then provides strategies to navigate through these models (and architectures) to select the best-performing one for a specific task and for running on a specific dataset. While building the NAS search space is already time and energy-consuming as well as computationally demanding, some strategies (Xie et al., 2023; Elsken et al., 2019; Zhao et al., 2021; Bender et al., 2018; Abdelfattah et al., 2021; Lin et al., 2021) require to train again few architectures using the new dataset to process so that they can predict performances.

To reduce such costs, an interesting approach is estimating performance without training, solely based on the topology and characteristics of the model (*e.g.,* (Xie et al., 2023)). We offer two performance predictors and compare them against a state-of-the-art one.

---

[3]https://codecarbon.io/

## 5.1 Image-Based Performance Prediction using ResNet18

Our first predictor simply builds a relation between the graphical representation of an architecture (*i.e.,* a PNG image generated with a call to the Keras function *plot_model* [4]) and its performance. The goal is to analyze the succession of layers (and their parameters) of an architecture and predict its performance. We chose a ResNet-18 model (He et al., 2016) as our prediction model since it is able to perform image processing (since the input is a PNG image) at a low cost.

## 5.2 Code-Based Performance Prediction using Transformers

Our second predictor relies on information contained in the actual Python code that is based on the description of an architecture stated in our DSL. This predictor is able to parse the code and focus on the different layers composing the architecture as well as extracting their hyperparameters. The goal here is to learn correlations between the different layers and their hyperparameters on the one hand and the performance of the architecture on the other hand. These are converted into a sequence of tokens that is the input of a Transformer model (Vaswani et al., 2017) designed for sequence-to-sequence tasks.

## 5.3 State-of-the-Art Performance Predictor

As said before, there are multiple recent methods that try to train a few architectures as a starting point for reasoning and selecting efficiently a suitable architecture (Xie et al., 2023). These methods are divided into different categories (such as few-shot learning or zero-shot learning) but they often use complex algorithms and heuristics to search for a suitable architecture. Neural Predictor for Neural Architecture Search (Wen et al., 2019), on the contrary, has decided to build a simple predictor (*i.e.,* a regression model). It is considered a few-shot learning method (see Table 1 in (Xie et al., 2023)) and showed good results in predicting (near-)optimal architectures at a lower cost. We found its implementation available and took it as our baseline.

# 6 Experiments

We now evaluate CNNGen by answering the following research questions:

**RQ1:** Can CNNGen generate architectures with a wider range of performances than those produced by NASBench101?

**RQ2:** How do our performance predictors compare to the state of-the-art?

The *first* question aims to verify that the DSL-based approach can generate a more diverse set of CNN architectures in terms of both performance and topology compared to a cell-based approach like NASBench101. Because of this, we claim it is more relevant to consider reasoning over our generator than over cell-based benchmarks and provide strategies to optimize the choice of CNN architectures to train.

The *second* question compares a performance predictor that we have selected from the literature with the ones that we presented in Section 5. In this evaluation, we do not aim to demonstrate that the two predictors we propose are better than the one mentioned in the literature. Rather, if they can perform similarly in their predictions, it could be worth considering other kinds of predictors that can rely on different artefacts than an abstract representation of the models.

## 6.1 Experimental Settings

To conduct our experiments, we randomly drew $1,300$ architectures from CNNGen. We used the popular CIFAR-10 dataset to train these models with 36 epochs. The whole dataset was split into $80\%$ for the training and $20\%$ for the test set. In the training set, $20\%$ of the data were used for validation. Once these models are trained, they are used for training the three considered performance predictors (see Section 5). The $1,300$ models were split into $80\%$ and $20\%$ respectively for

---

[4] https://keras.io/api/utils/model_plotting_utils/

the training and the test set. Again, $10\%$ of the training was dedicated to validation. To mitigate the random effect of this split, the training and evaluation of the predictors were repeated 10 times.

## 6.2 RQ1: CNNGEN VS NASBENCH101 REGARDING PERFORMANCE DISPERSION

To answer this question, we rely on the distribution of accuracy measures. We compare the distribution of these accuracy measures from the $1,300$ CNNGen architectures and from the $423,000$ from NASBench101 when evaluated on the test set. These distributions are depicted Figure 1a for NASBench101 and Figure 1b for CNNGen. As we mentioned previously, accuracy measures coming from NASBench101 architectures revolve mainly between $80\%$ and $85\%$. On the other hand, CNNGen architectures mainly provide accuracy measures between $40\%$ and $68\%$.

**Discussion.** CNNGen generates diversely performing random architectures. As it does not vary models against a reference one and does not optimize the search, it achieves a lower average performance for generated models. We see this as an opportunity to learn from bad models as well (and how to avoid wasting resources in their training) and produce balanced datasets. We state that this performance diversity stems from topology diversity. NASBench101 topologies can only vary inside the cells (3 stacks of 3 cells) with a maximum of 9 nodes and 7 edges. While NASBench101 authors enumerated all the possibilities, this is not possible with CNNGen due to its much larger space. Among our $1,300$ topologies, we were able to draw some with as few as 9 layers while others were about 272 layers. In comparison, supposing that all the cells would have 9 layers (*i.e.,* nodes), a stack would have a maximum of $9 \times 3 = 27$ nodes, and all three stacks would have $27 \times 3 = 81$ layers. Adding, the remaining layers would result in approximately a maximum of 85 layers. The smallest model coming from NASBench101 is about 15 layers.

In addition to performance, these diverse topologies showed diverse results in other metrics as well, such as energy consumption. Figure 5 presents the distribution of the energy consumption for the models generated with CNNGen. While we cannot compare with NASBench101 because it does not provide information about energy consumption, we can see some disparities amongst the models. Further investigations are needed to understand the possible relation between energy consumption and the number of layers in a model. It could also motivate the search for models that perform well (*e.g.,* above $80\%$ accuracy) while consuming a minimum of energy.

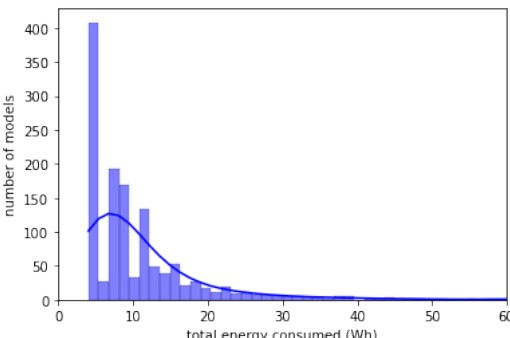

Figure 5: Disitribution of the energy consumption (in Watt-hour) measured via Code-Carbon for the $1,300$ models generated via CNNGen

---

**Conclusion (RQ1)**

With only $1,300$ randomly chosen models, we observe that the number of layers in topologies coming from CNNGen may be more diverse from 9 to 270 compared with 15 to 85 regarding NASBench101 for a much greater number of architectures. Our generated models with CNNGen report lower accuracies than the ones reported by NASBench101 but that could offer new opportunities for optimization search. The energy distribution shows promise in designing energy prediction models.

---

## 6.3 RQ2: PERFORMANCE PREDICTION

The goal is to compare whether our two predictors perform similarly with regard to the state-of-the-art predictor (*i.e.,* neural predictor). To answer this question we use the *Kendall rank correlation coefficient* (known as Kendall's tau, Ktau, or K$\tau$) and the *mean average error* (MAE) regarding the difference between the prediction and the observed accuracy measures. The first measure will ensure that, regardless of how far the predictions can be from actual accuracy measures, the predictors are still able to find the best-performing models. In fact, it guarantees that the ranking is respected. Ktau is defined between $-1$ and $1$ with $0$ stating that the ranking provided by the predictions is not correlated with the ones from the observed measures. The MAE will measure the mean error between observed measures and the predictions. It is defined between $0$ and $+\infty$, with $0$ a perfect prediction that never differs from observed measures. In addition, we perform a Mann-Whitney statistical test and report the p-value to support our observations. The null hypothesis is that the two sets of measures come from the same distribution. We consider that the null hypothesis can be refuted if the reported p-value is lower than $0.05$ (or $5e^{-2}$). Figures 6 and 7 report respectively the boxplots of ktau and MAE measures over $10$ repetitions when trying to predict based on the PNG image (called img, on the left), the Python code (called py_code, in the middle) and the neural predictor (called neural_predictor, on the right).

Figure 6 shows that the neural predictor has a mean K$\tau$ of $0.10$ which suggests that there is a low correlation between the ranking performed by the predictions and the ones with actual measures. We note that the *img* predictor provides a lower mean K$\tau$ (*i.e.,* $0.05$). On the contrary, the *py_code* predictor gets a higher mean K$\tau$ (about $0.30$). Thus, the *py_code* seems to be the best predictor among the three while the two others are comparable. This conclusion is supported by the upper part of Table 1. We can refute the null hypothesis when comparing *py_code* to either of the two other predictors. When comparing the distributions between *img* and *neural_predictor*, the reported p-value is closer to the limit (*i.e.,* $0.05$) but still below ($0.037$) which suggests that the neural predictor and its higher k$\tau$ coefficients should be considered second. Now, when considering the MAE and Figure 7, the mean value regarding *py_code* is still below than others. To be noted, there is an outlier in the distribution. Then, *img* has the second lower mean MAE and finally the neural predictor. Looking at the lower part of Table 1, the p-values reveal that *img* and *py_code* boxplots are different from the one from the neural predictor. However, with a p-value of $0.088$ when comparing the distributions coming from *img* and *py_code*, we cannot refute the null hypothesis.

**Discussion.** Providing the code of the generated models allows us to derive more accurate performance predictors and the image allows simple models to be on par with the state of the art. This demonstrates the relevance of providing more information than the adjacency matrix of the network to derive performance predictors.

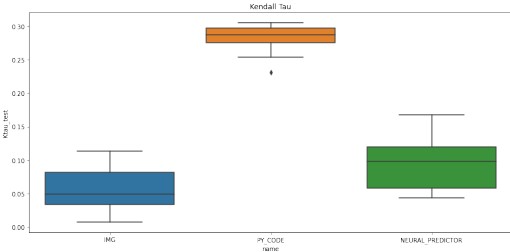

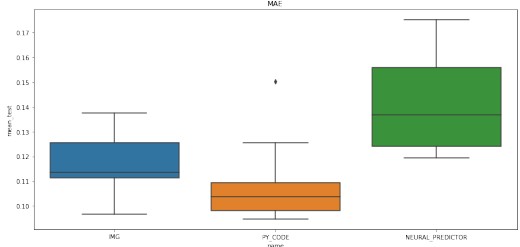

Figure 6: Boxplots reporting Kendall's tau (K$\tau$) from our 10 runs to compare the three different predictors. On the left, only the PNG image is used to predict, in the middle, the Python code is used, and on the right is the state-of-the-art predictor (*i.e.,* neural predictor).

Figure 7: Boxplots reporting mean average error (MAE) from our 10 runs to compare the three different predictors. On the left, only the PNG image is used to predict, in the middle, the Python code is used, and on the right is the state-of-the-art predictor (*i.e.,* neural predictor).

|  | P-value |
|---|---|
| **K$\tau$** | |
| img vs. py_code | $1.8e^{-4}$ |
| img vs. neural | $3.7e^{-2}$ |
| py_code vs. neural | $1.8e^{-4}$ |
| **MAE** | |
| img vs. py_code | $8.8e^{-2}$ |
| img vs. neural | $7.2e^{-3}$ |
| py_code vs. neural | $2.8e^{-3}$ |

Table 1: p-values Mann-Whitney statistical tests when comparing the results from the predictors against each others. The top part reports the p-values when comparing the K$\tau$ ranking correlation coefficient while the bottom part considers measures from the MAE.

> **Conclusion (RQ2)**
>
> Considering both measures (*i.e.,* k$\tau$ and MAE), our py_code is better on both measures than the state-of-the-art predictor when trying to predict the performance over our $1,300$ CNNGen models. Then, since *img* seems better than the neural predictor over MAE and vice-versa over K$\tau$ we cannot really conclude whether one prevails but rather we consider them equivalent.

## 7 CONCLUSION

This work introduced CNNGen, a context-free grammar-based architecture generator capable of producing a diverse range of CNN architectures tailored for image classification tasks within Neural Architecture Search (NAS). CNNGen also offers two new performance predictors. To evaluate CNNGen, we randomly generated $1,300$ architectures and observed their diversity in terms of topologies, performance or energy consumption. This diversity is much higher than for cell-based approaches such as NASBench101. We also evaluated our performance predictors against the state-of-the-art and demonstrated the utility of providing the model code and the image representation to build quality predictors. In particular, our code-based transformer performs best.

For future work, two research directions emerge. First, we would like to include NAS optimization techniques in CNNGen. For instance, Genetic Programming (GP) or Genetic Improvement (GI) techniques (Kifetew et al., 2017) can lead to computationally efficient explorations (Liu et al., 2022). GI would allow for automatic exploration and optimization of the architectures following an evolutionary strategy. The second research direction concerns the predictor design. We would like to explore prediction models for other quantities than performance, and in particular energy consumption or environmental impact. Our goal is to use such predictors during search to estimate accurately the property of a candidate network without having to train it. CNNGen paves the way for more sustainable NAS practices.

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

## A  CNNGEN SAMPLE ARCHITECTURES

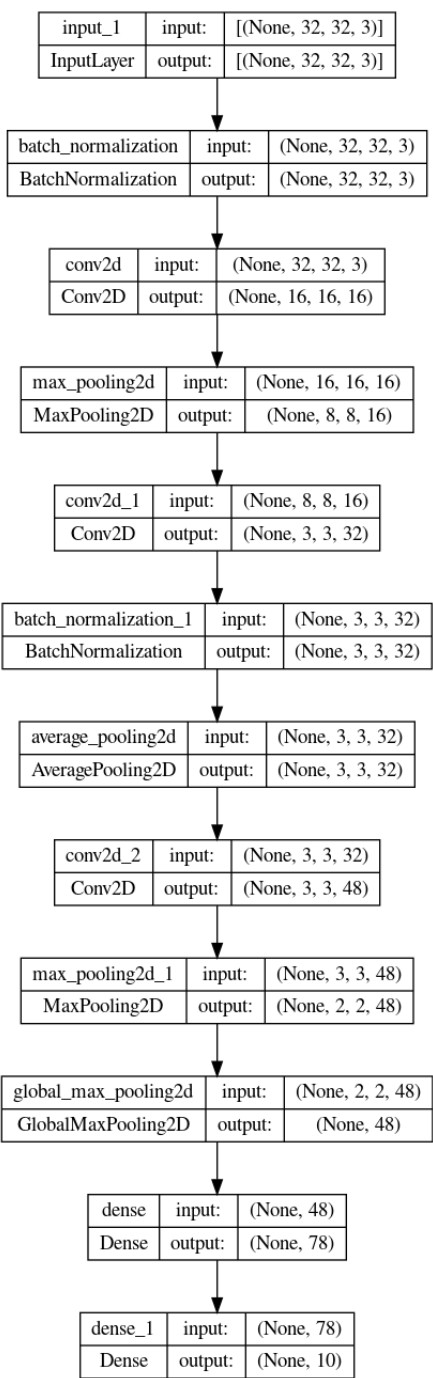

Figure 8: a visual representation of a CNN generated by CNNGen stored as a PNG image. This is used by one of the proposed performance predictors as input to predict the accuracy of a new CNN architecture (before training).

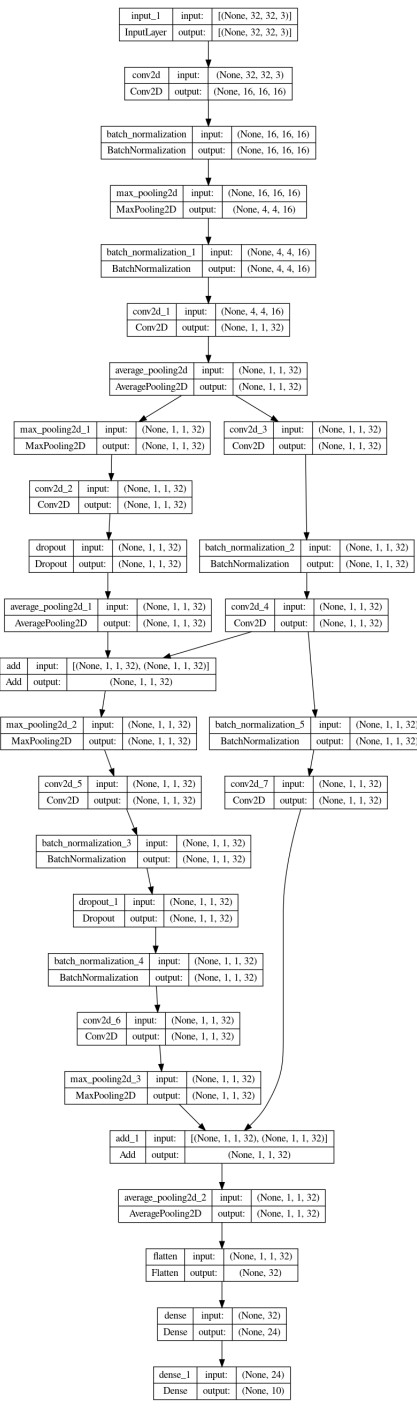

Figure 9: a visual representation of a CNN generated by CNNGen stored as a PNG image. This is used by one of the proposed performance predictors as input to predict the accuracy of a new CNN architecture (before training).

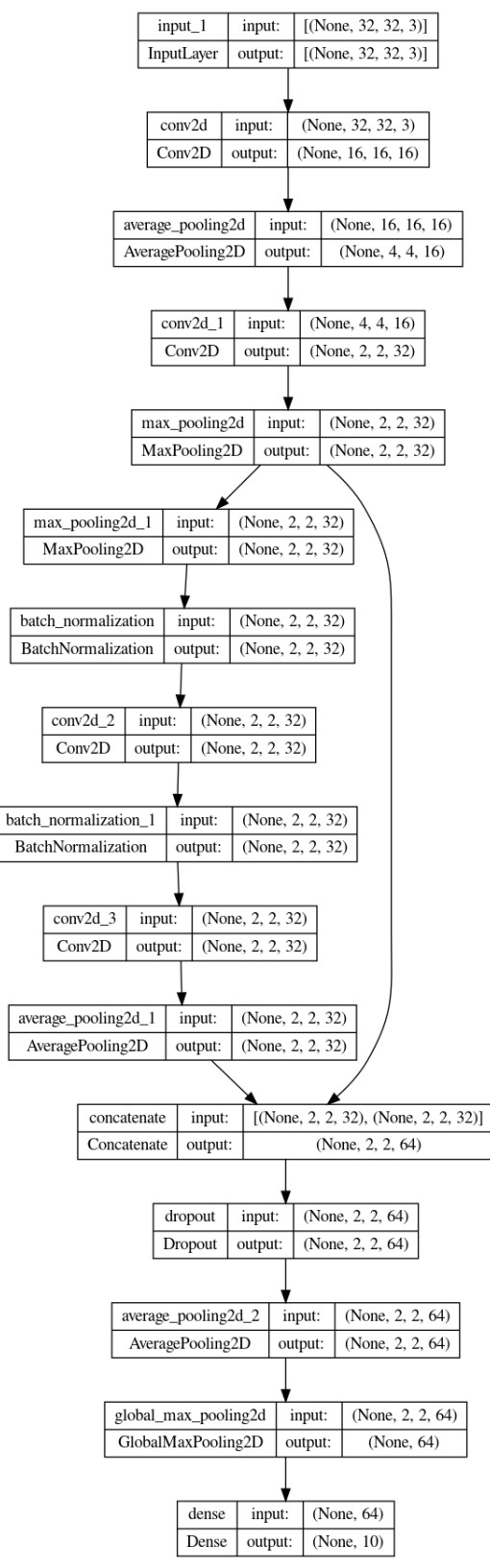

Figure 10: a visual representation of a CNN generated by CNNGen stored as a PNG image. This is used by one of the proposed performance predictors as input to predict the accuracy of a new CNN architecture (before training).

## B  CNNGEN'S GRAMMAR

| | | |
|---|---|---|
| CNNDescLang | : | cnndesclang = (Architecture); |
| Architecture | : | input = ('input')  fe += (FeatureExtraction)+  (inter = (Interstice)  class += (Classification)+)?  output = ('output'); |
| Dropout | : | 'dropout'; |
| Pooling | : | 'avg_pooling' \| 'max_pooling'; |
| Convolution | : | (bnconv = 'bnconv')  \|  (convbn = 'convbn')  \|  (conv = 'conv'); |
| GlobalPooling | : | ('global_avg_pooling') \| ("global_max_pooling"); |
| FlattenOrGlobal | : | (flat='flatten') \| (gp = GlobalPooling); |
| Interstice | : | fg = (FlattenOrGlobal); |
| Classification | : | drop = (Dropout)?  d=('dense'); |
| ConvDrop | : | conv = Convolution  (drop = Dropout)?; |
| MergeConv | : | merge=Merge  convdrop += (ConvDrop)*; |
| ConvOrMerge | : | convdrop += (ConvDrop)+  \|  convdrop += (ConvDrop)* mergeConv += MergeConv+; |
| Left | : | (p=Pooling)?  com=ConvOrMerge  (pool=Pooling)?; |
| Right | : | conv += Convolution+ \| {Right}  empty = 'Empty'; |
| MergeBody | : | '('  left = Left  virg=','  right = Right  ')'; |
| Merge | : | db = '['  (mergeBody+=MergeBody)+  fm = ']'; |
| FeatureExtraction | : | (conv = Convolution \| merge = (Merge))  drop = (Dropout)? pool = (Pooling)?; |

Figure 11: Complete CNNGen grammar rules

