# OpenReview forum: "CNNGEN: A GENERATOR AND BENCHMARK FOR SUSTAINABLE CONVOLUTIONAL NEURAL NETWORK SEARCH"
_ICLR.cc/2024/Conference — Submitted to ICLR 2024_

### Official Review · Reviewer_pJLK · 2023-10-31

**Soundness:** 1 poor
**Presentation:** 2 fair
**Contribution:** 1 poor
**Rating:** 3
**Confidence:** 2

**Summary:**

This work presents a new benchmark consisting of various performed architectures automatically generated from domain-specific language (DSL) and the performance, training code, and energy consumption information of each architecture.

**Strengths:**

This work presents a new benchmark and an image-based performance predictor.

**Weaknesses:**

1. Why the performance of networks should be diverse? Note the goal of NAS is to find architecture better than human crafted ones, while human designed architecture perform quite well in nearly all cases. The importance of NAS to distinguish models with similar high performance. For instance, NAS is designed to distinguish ConvNeXt and ResNeXt, rather than ConvNeXt and LeNet. The previous benchmark that has most architectures with very close and high accuracy actually make more sense than the proposed ones.

2. DSL has limitation in the expressivity of the architecture. The proposed DSL seems only supports sequential architecture without branches ( Fig 3 and Fig. 8).

3. Intuition of predictor. Why do you think a CNN can know the network's performance by looking at the image of its architecture? What is the intuition behind it? For me, it just overfits a small dataset.

**Questions:**

1. Why do we need an accurate number of energy consumption? Why not just model parameters and FLOPs? Energy consumption is very sensitive to the setup of machines and can be easily outdated. A model that consumes many energy might be very energe-saving in the next year due to the new software support and hardware update.

---

> ### Author Response · Authors · 2023-11-17
> **Response to Reviewer pJLK**
>
> Thank you for your review.
>
> Regarding weakness 1, we refer to our comment to reviewer (vffX): we want to evaluate and design predictors in a general way: these predictors should also be able to rule out bad architectures, which is not the case for the NAS benchmarks which all the topologies have similar good performance.  We agree intending to design (in the long term) better architectures than human-crafted ones.  Our motivation is that it extends to the whole topology and not only to parts (cells) of a given topology (i.e., architecture like LeNet). Therefore, our approach allows us to explore the CNNs’ search space in the large rather than in the small for a predefined topology.
>
>
> Regarding weakness 2, this is not correct. Our grammar can generate architectures with an infinity of branches in theory. In practice,  we constrain the topological space with at most 2 branches per node for a given network. It is true that we only showed linear ones in the original paper for the sake of simplicity, our mistake.  We have added in Appendix A a few more complex architectures generated in our benchmark. You can also see in the Zenodo in the folder output/architecure_img 1300 architectures that our generator synthesizes.
>
> Regarding weakness 3, the intuition behind using CNN to predict the network performance is that the image contains the topology and hyperparameters and this information is necessary to predict performance.
>
> We trained each predictor 10 times (see Section 6.1) and reported an average of the performance (cf code in Zenodo). There is also a CSV file that contains the results of each run.
>
> Regarding question 1, it is common to report energy consumption as it characterizes the environmental impact of these models. We did not see these concerns widely covered in existing benchmarks, which motivated providing a new one (in addition to network diversity). We do agree that it varies depending on hardware and software optimization. We rely on the code carbon library, which reports operations as well and could help design novel predictors in the future.

---

### Official Review · Reviewer_vffX · 2023-11-04

**Soundness:** 1 poor
**Presentation:** 3 good
**Contribution:** 1 poor
**Rating:** 3
**Confidence:** 4

**Summary:**

This paper presents a new way to generate a NAS search space. The authors dubbed their method CNNGen. This new methods relies on domain-specific language (DSL) to capture neural architectures using a dedicated grammar. The authors claim this methods results in more diverse architectures to search. The authors also present a method to compare their work against other search spaces and claim their method outperforms state-of-the-art. The authors also propose adding a carbon footprint metric to the models.

**Strengths:**

I highlight the following strengths:
- The idea of adding DSL to generate network candidates is interesting. Using grammar to represent networks is a good idea and it has the potential to represent more diversity of solutions as the authors point out in their results
- Representing an a network candidate as an image is also interesting and it allows for metrics that are more aligned with computer vision tools. The authors demonstrate the use of these metrics in the paper

**Weaknesses:**

I see 2 critical flaws with this work:
- The benchmark's are insufficient. The authors only compare the data with the one presented in [1]. This comparison is limited to CIFAR data only. Even in [1] the authors present a comparison using ImageNet data. Additionally, the authors present a benchmark but don't really benchmark any NAS methods. To show good power of the benchmark the authors need to present results showing that NAS methods can benefit from this benchmark and find architectures with better accuracy and even better carbon footprint (since the authors present this as a metric). As it stands I don't see sufficient evidence of novelty or impact for this work. The authors should run a real benchmark of methods and present more extensive results.
- The authors completely ignore the advancements in differentiable NAS ([2,3,4,5]). In these works there is no need to use a search space and one can optimize an architecture using gradient decent. In fact, some works like [3] and [5] show how to do this and also add constraints for latency and power consumption. Given the existence of these methods, I don't see the value of creating a new search space benchmark. I ask the authors to clarify the lack of mention and comparison here and clarify any misunderstanding from my side.

Given these flaws, I don't yet see merit in the work presented, the authors need to provide a more extensive benchmark and compare against more efficient methods like [2,3,4,5].


[1] Wen, Wei, et al. "Neural predictor for neural architecture search." European Conference on computer vision. Cham: Springer International Publishing, 2020.

[2] Liu, Hanxiao, Karen Simonyan, and Yiming Yang. "DARTS: Differentiable Architecture Search." International Conference on Learning Representations. 2018.

[3] Wu, Bichen, et al. "Fbnet: Hardware-aware efficient convnet design via differentiable neural architecture search." Proceedings of the IEEE/CVF conference on computer vision and pattern recognition. 2019.

[4] Li, Guohao, et al. "Sgas: Sequential greedy architecture search." Proceedings of the IEEE/CVF Conference on Computer Vision and Pattern Recognition. 2020.

[5] Li, Guohao, et al. "LC-NAS: Latency constrained neural architecture search for point cloud networks." 2022 International Conference on 3D Vision (3DV). IEEE, 2022.

**Questions:**

See weaknesses.

---

> ### Author Response · Authors · 2023-11-17
> **Response to Reviewer vffX**
>
> Thank you for your review and the references, we indeed used the neural predictor [1] as a reference to compare the performance predictors we designed.
>
> The motivation to use our generator to create a benchmark is to allow to design more generalizable predictors (which should stem from the diversity of networks created) and also to allow for predictors that focus on or integrate CNNs’ energy consumption. To the best of our knowledge, we did not see this information in existing NAS benchmarks.

---

### Official Review · Reviewer_WMeX · 2023-11-06

**Soundness:** 3 good
**Presentation:** 3 good
**Contribution:** 2 fair
**Rating:** 5
**Confidence:** 3

**Summary:**

The paper introduces a new approach to Neural Architecture Search (NAS) that uses a domain-specific language (DSL) to generate convolutional neural networks without predefined cells or base skeletons named CNNGen.
This approach offers a more diverse and sustainable solution to NAS, addressing the limitations of cell-based methods.
The paper outlines the comprehensive pipeline used by CNNGen to store network descriptions and fully trained models, and discusses the growing concern for sustainability in neural network design and implementation. Overall, CNNGen is an innovative and promising tool for generating and benchmarking sustainable convolutional neural networks.

**Strengths:**

- The paper introduces a new approach to use a domain-specific (natural) language (DSL) to generate convolutional neural networks, that allows the exploration of diverse and potentially unknown topologies.
- The paper discusses the growing concern for sustainability in neural network design, also computes energy consumption and carbon impact for green machine learning endeavors.

**Weaknesses:**

- The extracted five key concepts used to describe architectures (architecture, featureExtraction, featureDescription, classification) are still limited, not sure why this would lead to better diversity in neural architectures.

**Questions:**

- It's unclear how the author choose the five key concepts used to describe architecture.
- I feel the method is just trying to use natural language to replace the tradtional symbolic respresentation in NAS search spaces. I'm sceptical of why the proposed method could generate more diverse architectures.

---

> ### Author Response · Authors · 2023-11-17
> **Response to Reviewer WMeX**
>
> Thank you for your review.
>
> Regarding your first question, the five concepts are only the top-level ones in our grammar that represent the different phases of processing in a CNN. They were found based on our analysis of the literature and the function of the different layers (or groups of layers) from usual DNNs. In addition, we want to highlight that, as described in Section 4.1, we decompose these concepts into other ones to describe the network’s functions (convolution, pooling, batchNormalisation, etc.). We added the full Xtext Grammar in Appendix B. It contains 16 terminals and 17 non-terminals.
>
> Regarding your second question, we do not use natural language but a formal context-free grammar defining the language of possible CNN architectures. Therefore, it is also a symbolic approach. We achieve diversity by assembling network functions sequentially while respecting grammar rules. Our generator achieves diversity as it does not start with a predefined topology but combines the network functions sequentially while respecting the grammar rules. Thus, it can form combinations of multiple architectures in one network or completely new ones.  This diversity results in the varied CNN accuracies as depicted in Figure 1b.

---

### Author Response · Authors · 2023-11-17
**Response to all Reviewers**

Dear reviewers,

Thank you very much for your feedback, in addition to the individual comments, please see the list of changes we made to the paper:

- We added the full specification of the grammar in Appendix B. It shows how we capture the space of convolutional neural networks in more detail.

- We added a few more examples of generated architectures in Appendix A. They show that our generator can generate non-linear ones.

Best Regards,

The authors.

---

### Meta-Review · Area_Chair_JZHi · 2023-12-06

**Metareview:**

The paper proposes the use of a domain specific language (DSL) to generate neural architectures in neural architecture search (NAS). The primary strengths are that this is a novel take on the NAS task and that it has the prospect to strip away the need to pre-define cells and architecture skeletons. Despite the consensus on the strengths, with which the AC agrees, concerns remain with respect to the claimed diversity of architectures, justification of the methodology, and limited experimentation. Although a more detailed specification of the grammar and further examples of generated architectures were included in an appendix, these points remain largely unaddressed by the rebuttal.
At present, the paper seems to require these additional improvements before being ready for publication. Specifically, as mentioned by reviewer vffX, it will be necessary to revise the work with more experiments beyond a simple comparison on CIFAR10 and take into account related advances in differentiable NAS in experimentation and methodology. In addition, it is recommended to provide a more rigorous justification of the claim that the suggested method leads to more diverse architectures.

**Justification For Why Not Higher Score:**

The raised main weaknesses by the reviewers remain unaddressed. Most importantly, the justification and evidence for the proposed method requires improvement and given that there are no theoretical claims, the empirical demonstration needs major extension beyond a single dataset with limited comparison to related works.

**Justification For Why Not Lower Score:**

N/A

---

### Decision · Program_Chairs · 2024-01-16

Reject